# Impact of Psoriasis and Hidradenitis Suppurativa in Pregnancy, a Systematic Review

**DOI:** 10.3390/jcm10245894

**Published:** 2021-12-15

**Authors:** Maria-Angeles Ferrer-Alcala, Manuel Sánchez-Díaz, Salvador Arias-Santiago, Alejandro Molina-Leyva

**Affiliations:** 1School of Medicine, University of Granada, 18016 Granada, Spain; mariferrer@correo.ugr.es; 2Dermatology Unit, Hospital Universitario Virgen de las Nieves, IBS Granada, 18002 Granada, Spain; manolo.94.sanchez@gmail.com (M.S.-D.); alejandromolinaleyva@gmail.com (A.M.-L.); 3Dermatology Department, School of Medicine, University of Granada, 18016 Granada, Spain

**Keywords:** hidradenitis suppurativa, psoriasis, pregnancy

## Abstract

Psoriasis and hidradenitis suppurativa (HS) are chronic inflammatory skin diseases that frequently develop in young women. The aim of this study is to evaluate how hidradenitis suppurativa and psoriasis impact women desiring to conceive, and their influence on fertility and gestation. A systematic review of articles dating from January 2015 to April 2021 was performed using the Scopus (Elsevier) database. The search terms were (psoriasis and (birth or pregnancy or fertility)) and ((hidradenitis suppurativa or acne inversa) and (birth or pregnancy or fertility)). The search was limited to human data. Systematic reviews, case reports, clinical practice guidelines, expert consensus and conference papers were excluded. The impact of HS on pregnancy includes an impaired desire for pregnancy, a decrease in fertility, the worsening of the disease during pregnancy and potential adverse events during pregnancy. Moreover, the pregnancy might imply a change on the treatment of HS. The impact of psoriasis on pregnancy includes a decrease in fertility, potential adverse events during pregnancy and an unpredictable evolution of the disease. Moreover, the pregnancy might imply a change on the treatment of psoriasis, although biologic therapies do not appear to increase the risk of adverse events. In conclusion, both HS and psoriasis have an impact on pregnancy. A decrease of fertility has been reported. Moreover, both diseases have an unpredictable evolution during pregnancy. Pregnant women who are under biologic therapy do not seem to have a higher rate of adverse events. Treatment of both conditions is usually halted during pregnancy since scientific evidence about their safety is not conclusive, or teratogenic risk has been proven.

## 1. Introduction

Psoriasis is a chronic, immune-mediated skin disease characterized by the appearance of erythematous, scaly plaques on the scalp and extensor areas [1]. Moreover, inflammation is not limited to the skin, favoring a systemic pro-inflammatory state that is associated with multiple comorbidities, such as overweight, hypertension, dyslipidemia and diabetes, which produce an increased cardiovascular risk and a higher risk of myocardial infarction and stroke [2]. Regarding the impact on the mental health, depression and suicide risk are more frequent in psoriasis patients when compared to general population [3,4]. It is estimated to affect 2.3% of the Spanish population with no gender differences [5].

On the other hand, hidradenitis suppurativa (HS) is an immune-mediated skin disease characterized by the appearance of inflammatory nodules, abscesses and cysts, which generate fistulous tracts that release a purulent and malodorous liquid [6,7,8]. Main symptoms include itching and pain, leading to a decrease in quality of life [9,10]. As in psoriasis, in HS there is a component of systemic inflammation that correlates with the severity of the disease and is associated, among others, with an increased cardiovascular risk [11]. The disease is more common in women and specifically affects a considerable proportion of women of childbearing age [12].

Given the chronic inflammatory nature of both disorders, various immunosuppressive and immunomodulatory drugs are commonly used in their treatment [13,14,15]. Moreover, although they can occur at any age, both are common in young adults, a stage of life that coincides with the fulfillment of individuals’ reproductive desires. Given that there is a potential interference of these diseases or their treatments on the pregnancy, the aim of this systematic review is to evaluate the potential impact of both diseases on women who wish to become pregnant, the influence of both diseases on fertility and the impact of them on pregnancy outcomes.

## 2. Materials and Methods

A systematic review was performed to assess the impact of psoriasis and HS on gestational desire, fertility and pregnancy.

### 2.1. Objectives and Research Questions

The objective of this systematic review was to answer the research questions: 1. Do HS or psoriasis affect gestational desire? 2. Do HS or psoriasis imply a decrease in fertility? 3. Do HS or psoriasis affect the evolution of pregnancy? 4. How do these diseases behave during pregnancy? 5. Do treatment of HS or psoriasis affect pregnancy? 6. Ds pregnancy entail a change in treatment?

### 2.2. Inclusion and Exclusion Criteria

The search included articles published in journals included in the Medline and Embase databases. The search was limited to: (a) human data, (b) articles in English and Spanish and (c) articles published from January 2015 to April 2021. All types of epidemiological studies on the impact of HS and psoriasis on gestational desire, fertility and pregnancy were included. Systematic reviews, case reports, clinical practice guidelines, expert consensus and conference papers were excluded.

### 2.3. Bibliographic Search and Data Extraction

Articles published from January 2015 to April 2021 were included, using the Scopus database (Elsevier), since this search engine includes the Medline and Embase repository. Two search algorithms were used, one for each disease: (*psoriasis and* (*birth or pregnancy or fertility*)) and ((*hidradenitis suppurativa or acne inversa*) *and* (*birth or pregnancy or fertility*)).

All the articles obtained in the initial search were assessed by title and abstract review to check the eligibility criteria. Articles meeting eligibility criteria were full-text reviewed. Finally, all the articles which met inclusion criteria were included in the study (Figure 1, Figure 2). All the articles were independently reviewed by two researchers (MAFA, MSD), if the two authors disagreed on the inclusion of a study, a third author (AML) reviewed the study until consensus was reached.

### 2.4. Variables

Variables included in the study were: Epidemiological study design, level of scientific evidence according to the Center for Evidence-Based Medicine, sample size and outcome variables in relation to the research question(s) that apply to each study. The levels of scientific evidence considered were as follow: 1a, systematic review or meta-analysis of randomized clinical trials; 1b, randomized clinical trial; 2a, systematic review or meta-analysis of cohort studies; 2b, cohort study; 2c, ecological trial; 3a, systematic review or meta-analysis of case-control studies; 3b, case-control study; and 4, case series and cohort and case-control studies of poor quality [16].

### 2.5. Risk of Bias Assessment

The risk of bias for the studies included in the review was assessed following the “Quality Assessment Tool for Observational Cohort and Cross-Sectional Studies” of the National Institutes of Health. The results of this evaluation can be seen in the Appendix A.

## 3. Results

The search on HS yielded 46 articles, of which seven answered the questions posed (Figure 1). All the studies had a level of evidence 4. Regarding psoriasis, 504 articles were obtained and 27 were chosen for answering the questions (Figure 2). The category of the articles included can be seen in Table 1.

### 3.1. Hidradenitis Suppurativa

#### 3.1.1. Does HS Affect Gestational Desire?

HS could involve or delay gestational desire fulfillment; however, the scientific evidence in this regard is scarce and of low quality. In the study by Adelekun AA et al. less than 17% (10/59) of women responding to an electronic survey indicated little interest in having children because of HS and more than 33.9% (20/59) planned to have more children in the future [13]. In the work of Montero-Vilchez et al. 50.96% (53/104) of women of childbearing age had an unsatisfied gestational desire. This group had an earlier onset of the disease and a higher educational level than women with a satisfied gestational desire (49.04%) [14].

#### 3.1.2. Does HS Imply a Decrease in Fertility?

One study evaluated decreased fertility in men and women with HS. In this study by D Tzur Bitan et al. the association was stronger in the 36–45 years age group with an odds ratio of 4.5 (2.55–7.93 95% CI) and in women relative to men with an odds ratio of 3.1 (2.57–3.74 95% CI) [15]. In the study by Adelekun AA et al. 8.47% (5/59) women with HS surveyed responded that they had fertility problems despite 12 months of trying to become pregnant, 11.86% (7/59) were on treatment to improve fertility, 45.76% (27/59) had been pregnant and 3.38% (2/59) were pregnant at the time of the survey [13].

#### 3.1.3. Does HS Negatively Affect Pregnancy Outcome?

Three articles evaluate the complications that can occur during pregnancy and pregnancy outcomes [13,16,17].

In the study by Adelekun A. A. et al., of 27 women with HS who had been pregnant, 74.07% (20/27) responded that they had had full-term children, 14.81% (4/27) suffered a miscarriage and 11.11% (3/27) underwent an abortion [13].

The study by Fernandez, J. M. et al. found that in women with HS with genital involvement who had given birth vaginally, 3.1% believed that the disease interfered with delivery and 23.5% believed that vaginal delivery caused a flare-up of the disease. Physicians advised cesarean section in 3.6% (10/279) of women for severe pubic involvement. A total of 33.9% (41/121) of women who underwent cesarean section indicate that HS interfered with incisional healing and 52.1% (63/121) with the development of incisional lesions [17].

In the work of Lyons AB et al. observed that greater severity as assessed by Hurley stages was not associated with a greater likelihood of miscarriage, nor was having vulvar and inguinal lesions associated with a greater likelihood of cesarean section. Initial disease severity was not associated with increased likelihood of cesarean section or increased risk of gestational hypertension, preeclampsia, neonatal or pregnancy complications. In a cohort of patients from the United States, a higher rate of gestational hypertension and preeclampsia was evident when comparing pregnancy complications with the general population [16].

#### 3.1.4. How Does HS Behave during Pregnancy?

Three studies answered this question [17,18,19]. According to these studies HS may worsen during pregnancy, although the risk cannot be determined exactly.

In the study by Vossen, A. R. R. J. V et al. 53.1% (*n* = 96) of women experienced no change in HS during pregnancy, in 30.2% (*n* = 96) HS symptomatology improved and 16.7% (*n* = 96) worsened. No association was found between Hurley stage and symptoms during pregnancy. Those who experienced worsening symptoms with menstruation were significantly more likely to improve during pregnancy (18). In the study by Fernandez, J. M. et al. HS symptoms worsened in 34.8% (97/279), improved in 36.6% (102/279) and were unchanged in 28.7% (80/279) [17]. Finally, in the study by Lyons AB et al., in 61.9% (70/113) the disease worsened, 7.9% (9/113) improved and in 30% (34/113) there was no change. Women in Hurley stage I worsened more than those in stage II or III. In addition, those with worsening disease were more likely to be receiving treatment for hidradenitis suppurativa [19].

#### 3.1.5. Does Treatment of HS Affect Pregnancy?

No studies have been found to answer this question.

#### 3.1.6. Does Pregnancy Mean a Change in The Treatment of HS?

It seems that pregnancy could mean a change in the treatment of HS. However, the scientific evidence is scarce. In the study by Lyons AB et al., a case series, 39.53% (17/43) of the patients discontinued treatment because they became pregnant. The reason was specified by 76.47% (13/17), 23.07% (3/13) by their own preference and 76.92% (10/13) by physician recommendation. Topical clindamycin was the most frequently discontinued treatment [19].

One of the objectives of the work of Montero-Vilchez et al. was to describe the treatments prescribed in women with HS who had fulfilled their gestational desire compared to those who had an unfulfilled gestational desire. It was observed that the frequency of prescription among women with fulfilled and unfulfilled gestational desire is similar. A decrease in the prescription of acitretin, which is teratogenic, was observed in women with unsatisfied gestational desire, although the association was not statistically significant [14].

### 3.2. Psoriasis

#### 3.2.1. Does Psoriasis Affect Gestational Desire?

No study has been found to answer this question.

#### 3.2.2. Does Psoriasis Lead to Decreased Fertility?

Six studies answer this question [20,21,22,23,24,25]. In the study by Gonzalez-Cantero et al. conducted in Spain, a 50% decrease in fertility rate is observed in women with moderate-severe psoriasis [25]. In the study by Lambe et al. the mean number of children at the end of the reproductive period did not differ between women with psoriasis and controls, but the proportion of nulliparous women was slightly lower in women with psoriasis at the end of the reproductive age [22]. In the study by Filippi et al. no decrease in fertility was observed in male patients treated with biologics [20].

Other studies show hormonal and reproductive parameter alterations in both men and women. In the study by Tuğrul Ayanoğlu et al. women with psoriasis have elevated FSH levels and an elevated FSH/LH ratio compared to healthy controls. In addition, they have a decreased antral follicle count. There is no correlation between the severity of psoriasis and the degree of involvement of these parameters [23].

Caldarola et al. compared hormone levels and seminal parameters of untreated males with psoriasis with healthy controls. He observed a decrease in testosterone and sex hormone-binding protein and an increase in estrogen. Sperm count, motility and percentage of sperm with normal morphology were decreased in patients with psoriasis, and half of the patients with psoriasis had one or more semen parameters altered. Signs of inflammation in the accessory glands were seen on ultrasound in 70% (35/50) of males with psoriasis [21].

Finally, in the study by Heppt et al. 14.8% (4/27) of men with psoriasis had all normal seminogram values, while 85.2% (23/27) had at least some alteration in some of the seminogram values and 8.69% (2/23) of them were diagnosed with azoospermia. In this sample 48.1% (13/27) of the patients presented elevated inflammation parameters (leucospermia and polymorphonuclear elastase) without clinical manifestation. In patients on systemic anti-inflammatory treatment with anti-TNF alpha or fumaric acid esters, several semen samples were determined and there was a decrease in inflammatory parameters and no changes were observed in the seminogram with respect to the first sample, although statistical significance was not reached due to the small sample size [24].

#### 3.2.3. Does Psoriasis Affect the Course of Pregnancy?

Eight articles answer this question [22,26,27,28,29,30,31,32]. There seems to be an increased risk of complications during pregnancy. Although there is some controversy among the different studies (Table 2).

In the study by H. Jin et al. a subtype of a rare variant of psoriasis, generalized pustular psoriasis of pregnancy, was associated with severe complications, with 50% (1/2) of cases resulting in maternal death and 50% (1/2) in miscarriage [28].

#### 3.2.4. How Does Psoriasis Behave during Pregnancy?

Seven studies answer this question [27,28,33,34,35,36,37]. The evolution of psoriasis during pregnancy is unpredictable and worsening is observed in all studies (Table 3). In the study by Maccari et al. 21% (*n* = 149) of pregnant women responded that their psoriasis worsened during pregnancy, in 34% (*n* = 149) it improved and in 45% (*n* = 149) it did not change [35]. In the study by J. M. E. Boggs et al. 66.7% (6/9) of women who stopped biologic treatment because of pregnancy had a severe flare-up during the course of pregnancy [33].

Two studies investigate the clinical and course of generalized pustular psoriasis, a rare variant of psoriasis. The study by H. Jin et al. 100% (2/2) of women with generalized pustular psoriasis of pregnancy, a subtype of generalized pustular psoriasis, developed first-time plaque psoriasis and pustular lesions during pregnancy [28]. In the study by Lau et al. it was observed that pregnancy was the trigger for 18.75% (3/16) of cases of acute generalized pustular psoriasis [35].

#### 3.2.5. Does Psoriasis Treatment Affect Pregnancy?

Fourteen studies answer this question [26,34,38,39,40,41,42,43,44,45,46,47,48,49]. The drugs studied in all the articles are biologic drugs. In one of them the study population was men and in the rest women. Based on the available scientific evidence it seems that treatment with biologic drugs during pregnancy is safe, in all the studies most of the pregnancies were carried to term, without complications and with healthy children (Table 3).

#### 3.2.6. Does Pregnancy Change the Treatment of Psoriasis?

Thirteen studies answer this question [35,38,39,40,41,42,43,44,45,46,47,48,50]. Pregnancy seems to cause a change in the treatment of psoriasis. With regard to biologic drugs, a significant proportion of patients discontinue treatment because they have become pregnant (Table 3).

In the study by Maccari et al. 37% (56/149) of patients stopped treatment during pregnancy. The most frequently withdrawn treatment was ultraviolet B (UVB) and ultraviolet-A (UVA) phototherapy (20%), methotrexate (18%), retinoids and cyclosporine (7% both) and biologics (3.6%). A total of 37.5% (*n* = 149) of the women stopped local treatments [27]. In the study by Belleudi et al. it was observed that in pregnancy there is a general withdrawal of all psoriasis treatments. Before pregnancy, 33.7% (177/525) of women had no treatment and this percentage increased to 76% of women in the first trimester of pregnancy, reaching 88.1% (458/520) in the third trimester. Before pregnancy, 52% (273/525) of the women used topical treatment, decreasing to 9.4% (49/520) in the third trimester of pregnancy. The use of systemic drugs also decreased during pregnancy, from 5.9% (31/525) use before pregnancy to 2.5% (13/520) in the third trimester of pregnancy [47].

## 4. Discussion

The results of this systematic review show that psoriasis and HS may lead to decreased fertility, may increase the number of adverse events during pregnancy and gestation may trigger a clinical worsening of the disease.

Gestational desire is influenced by multiple factors, among them is the diagnosis of a chronic disease that could delay or abolish that desire [48]. In HS patients with unsatisfied gestational desire, it is less common to be married and more common to be divorced [14]. Decreased gestational desire may not only be observed in HS, as it may occur in other chronic diseases affecting women of childbearing age. For example, in the study by Birgit S. Blomjous et al. one third of women with systemic lupus erythematosus reported not having become mothers because of their disease [49]. Similar studies have been reported for other systemic diseases [50].

The decreased fertility with which HS and psoriasis appear to be associated could exacerbate the increase in psychiatric comorbidities already prevalent in these patients [51]. Some of the factors that may interfere with fertility are sexual dysfunction, socioeconomic status or marital status. In both HS and psoriasis a higher incidence of sexual dysfunction has been reported [52,53]. It has also been observed that women with HS are more frequently unmarried [54].

Regarding the clinical course of both diseases during pregnancy, variable results have been observed. Pregnancy implies an increase in immune tolerance and an improvement of the disease could be expected; however, cases of worsening are also described, as in the case of pustular psoriasis of pregnancy [55]. In HS, in addition to the inflammatory component, there is a hormonal component that may also be modified during pregnancy. Consistently, in those cases in which psoriasis improved during pregnancy, it tends to be reproduced in subsequent pregnancies. Psoriasis has also been observed to improve with high doses of oral estrogen contraceptives [56]. Those women who were more likely to improve in pregnancy if their clinical condition worsened during menstruation could be due to progesterone levels, which decrease during menstruation and increase during pregnancy [57].

It has been observed that in moderate-severe psoriasis there is an increased risk of complications during pregnancy, this could be related to the systemic inflammatory imbalance that these patients present. In premature and small-for-gestational-age newborns born to mothers with psoriasis, there is evidence of increased proinflammatory cytokines in the umbilical cord [56]. However, in HS no correlation was observed between severity as determined by Hurley stages and an increased likelihood of complications. The increase in adverse outcomes during pregnancy that we observed in this review could also be due to the multiple comorbidities associated with psoriasis (diabetes mellitus, overweight, dyslipidemia…) [2]. HS is also associated with multiple comorbidities, such as other autoimmune diseases, obesity, diabetes mellitus and increased cardiovascular risk [11,58,59].

With regard to the treatment of psoriasis, it must be taken into account that some drugs are contraindicated during pregnancy because of their high teratogenic risk, such as retinoids and methotrexate. We observed that most women withdraw their treatment with biologic drugs when they become pregnant, and this is because the Food and Drug Administration determines that their use should be avoided during pregnancy due to insufficient evidence on their safety (Category B). We have also observed the withdrawal of other drugs, such as cyclosporine and psoralen-ultraviolet A (PUVA), which belong to category C and whose use should be done on a benefit/risk basis, since they have demonstrated adverse effects in animal studies [60,61,62,63]. Except for certolizumab pegol, anti-TNF alpha drugs are monoclonal antibodies that can cross the placental barrier as early as 22 weeks of gestation, and for this reason it is recommended that they be discontinued in the third trimester of pregnancy. The reason why certolizumab hardly crosses the placental barrier is that its structure lacks the Fc fraction [64,65,66,67].

In the treatment of HS the same occurs as in psoriasis, we observe a withdrawal of treatment during pregnancy because these are drugs for which their safety has not been demonstrated or they are contraindicated. Acitretin belongs to category X. Within category D we find systemic corticosteroids and tetracyclines and in category C, intralesional corticosteroids. Topical clindamycin and adalimumab belong to category B [61,62,68].

Finally, it should be taken into account that HS patients have a poor self-image and poor self-confidence, and therefore it is usual that they do not start relationships [69].

The results of this systematic review should be considered taking into account the presence of a series of methodological limitations. The main one is the low level of evidence of the available articles, since pregnancy or unsatisfied gestational desire are exclusion criteria or preclude participation in clinical trials, which constitute the highest level of scientific evidence. However, given the great relevance of this topic and its potential impact on health, it is essential to carry out quality observational studies.

## 5. Conclusions

The results of this review show that psoriasis and HS seem to impact different reproductive aspects of patients (Figure 3). On the one hand, HS may decrease gestational desire and both entities have been related to decreased fertility. Complications during pregnancy in women with psoriasis appear to be more frequent than in the general population. The evidence regarding HS is very scarce. The course of both diseases during pregnancy is unpredictable. In pregnant women who maintain treatment with biologic drugs, a higher rate of adverse effects is not observed. Considering this fact and the fact that psoriasis can worsen during pregnancy, the use of biologic drugs in moderate-severe cases could be justified.

## Figures and Tables

**Figure 1 jcm-10-05894-f001:**
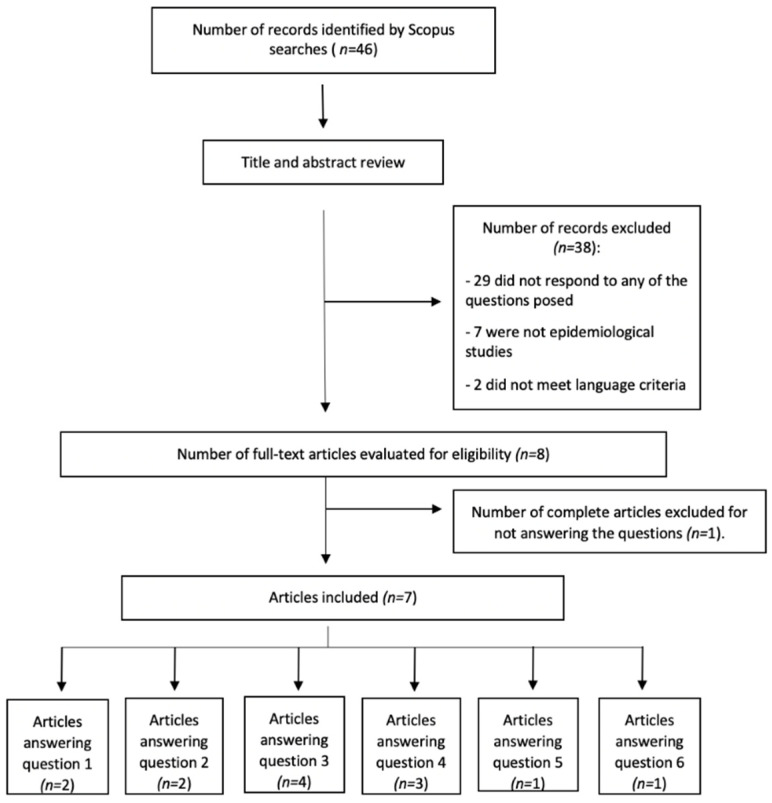
Search strategy, hidradenitis suppurativa.

**Figure 2 jcm-10-05894-f002:**
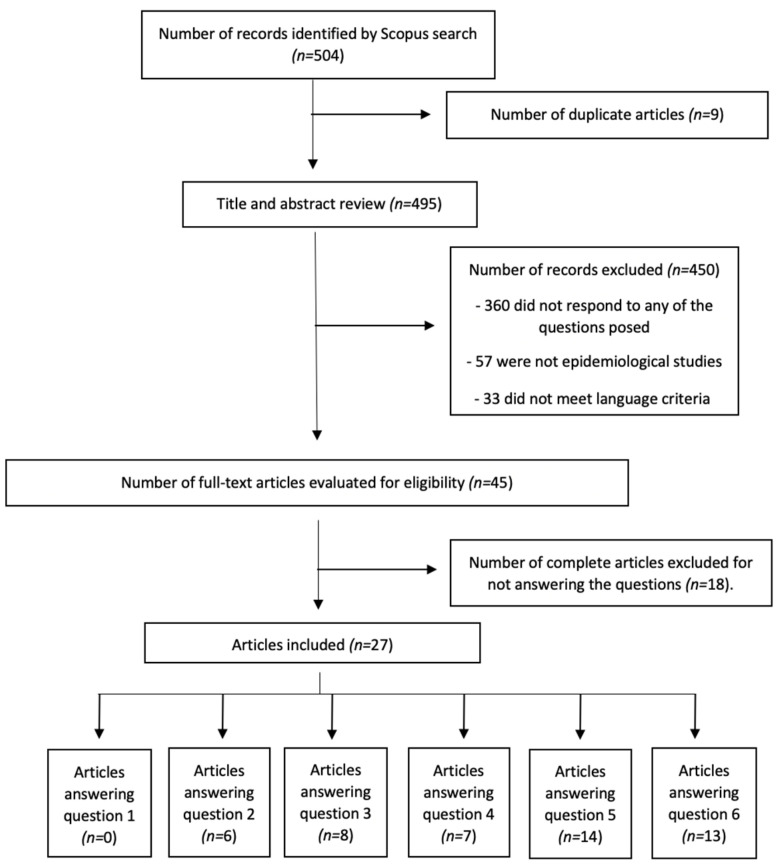
Search strategy, psoriasis.

**Figure 3 jcm-10-05894-f003:**
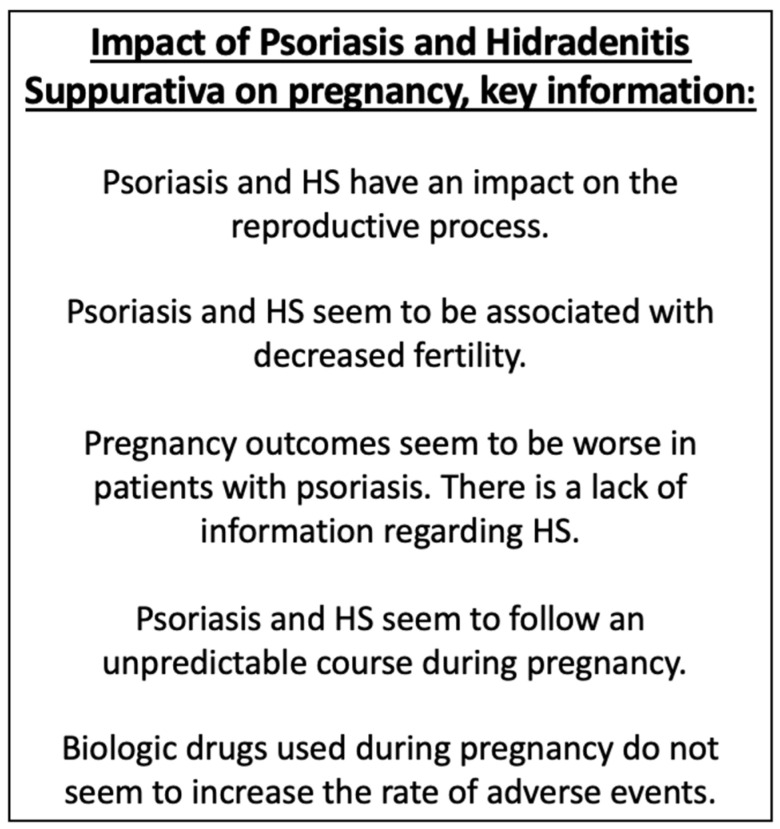
Key information.

**Table 1 jcm-10-05894-t001:** Main characteristics of the studies included in the systematic review.

Study	Study Design	Number of Patients	Control Population ^a^	Question Which Is Answered	CEBM
Hidradenitis Suppurativa					
Adelekun A. A. et al. [13], 2020, United States	Cross-sectional	59	-	1, 2, 3	4
Montero-Vilchez et al. [14], 2021; Spain	Case series	104	-	1, 6	4
D. Tzur Bitan et al. [15], 2021; Israel	Cross-sectional	4191	20,941	2	4
Vossen, A. R. J. V et al. [18], 2017; Netherlands	Cross-sectional	96	-	4	4
Fernández, J. M. et al. [17], 2021; United States	Cross-sectional	279	-	3, 4	4
Lyons A. B. et al. [16], 2020; United States	Case series	127	-	3	4
Lyons A. B. et al. [19], 2020; United States	Case series	127	-	4, 6	4
Psoriasis					
González-Cantero et al. [25], 2019; Spain	Prospective cohort study	732	-	2	2b
Lambe et al. 1 [22], 2020; Sweden	Cross-sectional	33,488	1,634,095	2	4
Lambe et al. 2 [22], 2020; Sweden	Population cohort study	15,975	1,464,517	3	2b
Filippi et al. [20], 2020; Italy	Case series	32	-	2, 5	4
Tuğrul Ayanoğlu et al. [23], 2018; Turkey and United States	Prospective case-control study.	14	35	2, 3	3b
Caldarola et al. [21], 2017; Italy	Case-control study.	50	50	2	3b
Heppt et al. [24], 2017; Germany	Case series	27	-	2	4
Bandoli & Chambers [26], 2017; United States	Case-control study.	330	1703	3	3b
G. Bröms et al. [31], 2018; Denmark and Sweden	Population cohort study	8097	943,846	3	2b
Bandoli et al. [29], 2020; United States	Population cohort study	1255	2,962,633	3	2b
Y. Huang et al. [30], 2021; Taiwn	Population cohort study	4058	2,346,281	3	2b
Kimball et al. [32], 2021; United States	Cohort study	220	-	3, 5	2b
Maccari et al. [27], 2020; France	Cross-sectional	149	-	3, 4, 6	4
H. Jin et al. [28], 2015; Corea	Case series	2	-	3, 4	4
Lau et al. [35], 2017; Malaysia	Case series	27	-	4	4
J. M. E. Boggs et al. [33], 2020; Ireland	Case series	8	-	4, 5, 6	4
Echeverría-García et al. [34], 2017; Spain	Case series	7	-	4, 5, 6	4
Galluzzo et al. [37], 2019; Italy	Case series	3	-	4, 5, 6	4
Odorici et al. [36], 2019; Italy	Case series	12	-	4, 5, 6	4
Kawai et al. [42], 2019; Japan	Case series	73	-	5, 6	4
Watson et al. [44], 2019; United Kingdom	Case series	7	-	5, 6	4
Lund & Thomsen [45], 2017; Denmark	Case series	7	-	5, 6	4
Haycraft et al. [38], 2020; United States	Case series	14	-	5, 6	4
M. E. B. Clowse et al. [39], 2016; different countries	Case series	16	-	5, 6	4
Babuna Kobaner and Polat Ekinci [40], 2020; Turkey	Case series	6	-	5, 6	4
Russo et al. [41], 2021; Italy	Case series	4	-	5, 6	4
Carman et al. [43], 2017; United States ^b^	Retrospective cohort study	81	1754	5	2b
Belleudi et al. [47], 2020; Italy ^c^	Retrospective cohort study	525	293	6	2b

^a^ If not otherwise indicated, control group is conformed by healthy participants. ^b^ The case group consists of patients with psoriasis treated with etanercetp and the control group consists of patients with psoriasis not exposed to etanercept or other anti-TNF alpha and patients without psoriasis. ^c^ The case group consists of women with psoriasis and the control group consists of women with rheumatoid arthritis. Abbreviations: CEBM, Center for Evidence-Based Medicine. 1a: Evidence obtained of systematic reviews or meta-analysis of randomized control trials; 1b: Evidence obtained from individual randomized control trials; 2a: Evidence obtained from systematic reviews or meta-analysis of cohort studies; 2b: Obtained from individual cohort studies; 3a: Obtained from systematic reviews or meta-analysis of case-control studies; 3b: Obtained from individual case-control studes; 4: Obtained from case series; and 5: Obtained from expert opinions.

**Table 2 jcm-10-05894-t002:** Articles addressing the issue of psoriasis and pregnancy.

	Lambe et al. [22], 2020 ^a^	Bandoli & Chambers [26], 2017	G. Bröms et al. [31], 2018 ^b^	Bandoli et al. [29], 2020	Y. Huang et al. [30], 2021 ^c^	Kimball et al. [32], 2021	Maccari et al. [27], 2020
Preterm delivery	OR = 1.07 (0.99–1.15)	14.2% (47/330) ^d^	OR = 0.97 (0.87–1.08)	10.8% (135/1255)	OR = 1.13 (1.02–1.25)	9.1% (22/243)	-
Gestation at term	-	-	-	-	-	90.9% (221/243)	97.3% (145/149)
Preeclampsia	OR = 1.09 (0.99–1.19)	6.4% (21/330) ^e^	OR = 1.15 (1.01–1.30)	-	OR = 1.57 (1.31–1.89)	-	-
Gestational hypertension	OR = 1.37 (1.19–1.58)		OR = 1.17 (1.02–1.35)	-	OR = 1.42 (1.12–1.80	-	-
Gestational diabetes	-	10% (33/330)	OR = 1.20 (1.02–1.40)	-	OR = 1.13 (1.00–1.27)	-	-
Newborn small for gestational age	OR = 1.00 (0.90–1.11)	-	OR = 0.95 (0.84–1.09)	7.8% (98/1255) ^g^	OR = 1.12 (1.02–1.23)	-	-
Newborn large for gestational age	OR = 1.11 (1.01–1.21)	-	-	-	OR = 0.97 (0.88–1.07)	-	-
premature rupture of membranes	OR= 1.15; (1.04–1.27)	-	-	-	-	-	-
Congenital malformations	Cleft palate ((OR = 1.69); 1.07–2.66) and non-specified malformations (OR, 1.08; 1.01–1.16)	-	OR = 1.01 (0.90–1.13)	-	OR = 0.87 (0.75–1.00)	0.8% (2/244)	-
Cesarean section	-	-	OR = 1.11 (1.02–1.20) ^f^	42.1% (528/1255)	OR = 1.06 (1.01–1.12)	-	-
Antepartum hemorrhage	-	-	OR = 1.04 (0.90–1.19)	-	OR = 0.98 (0.84–1.13)	-	-
Postpartum hemorrhage	-	-	-	-	Uterine atony: OR 1.41 (1.01–1.98) Severe hemorrhage: OR 1.57 (1.36–1.82)	-	-
Newborn stillborn	OR = 1.23 (0.96–1.57)	-	OR = 0.74 (0.46–1.19)	-	OR = 1.48 (1.11–1.96) ^g^	0.4% (1/244)	-

^a^—OR estimated from multivariable logistic regression adjusted for maternal smoking habits (during the first trimester), age at delivery, pre-pregnancy BMI and delivery period. Adjusted for intra-sister correlation using a robust standard error estimator. ^b^—OR adjusted for country, maternal age, parity, body mass index, hypertension, diabetes, mellitus, depression, smoking. ^c^—OR adjusted for age, child sex, urbanization, income, occupation, year of birth, Charlson comorbidity index and maternal nationality. Other adverse events associated with psoriasis in pregnancy include Apgar at 5 min (<7) OR = 1.53 (1.07–2.19) and low birth weight OR = 1.27 (1.14–1.41). ^d^—Women with psoriasis have an increased risk of preterm delivery. Reduction in gestation by an average of 0.30 weeks (95% CI −0.55–0.05) in the linear regression model. ^e^—The association between psoriasis and preeclampsia is high but not significant. ^f^—Elective cesarean section. Emergent cesarean section OR = 1.09 (1.01–1.18). ^g^—Unexplained stillbirth: OR = 1.61 (1.21–2.14).

**Table 3 jcm-10-05894-t003:** Characteristics of biological treatment during pregnancy and pregnancy outcomes.

Study	Treatment	Number of Pregnancies	Treatment Discontinuation ^a^	Spontaneous Abortion	Elective Abortion	Healthy Newborn ^b^	Complications ^b^
Filippi et al. [20]	Adalimumab	11	-	18.2% (2/11)	0% (0/11)	100% (9/9)	-
	Ustekinumab	6	-	0% (0/6)	0% (0/6)	100% (6/6)	-
	Etanercept	9	-	0% (0/9)	11.1% (1/9)	100% (8/8)	-
	Infliximab	15	-	0% (0/15)	20% (3/15)	100% (12/12)	-
	Secukinumab	3	-	0% (0/3)	0% (0/3)	100% (3/3)	-
Kimball et al. [32]	All biologic treatments	168	-	16.7% (28/168)	5.4% (9/168)	93.1% (122/131)	0.8% (1/131) Newborn stillborn 0.8% (1/131) Congenital malformations
	Ustekinumab	70	-	14.3% (10/70)	5.7% (4/70)	94.6% (53/56)	1.8% (1/56) Congenital malformations
	Infliximab o golimumab	14	-	7.1% (1/14)	0% (0/14)	92.9% (13/14)	-
	Other biologic treatments ^c^	84	-	20.2% (17/84)	6% (5/84)	90.3% (56/62)	1.6% (1/62) Newborn stillborn
J. M. E. Boggs et al. [33]	Adalimumab	9	55.6% (5/9)	11.1% (1/9)	0% (0/9)	100% (8/8)	-
	Infliximab	1	0% (0/1)	0% (0/1)	0% (0/1)	100% (1/1)	-
	Ustekinumab	1	0% (0/1)	0% (0/1)	0% (0/1)	100% (1/1)	-
	Etanercept	1	100% (1/1)	0% (0/1)	0% (0/1)	100% (1/1)	-
	Secukinumab	2	50% (1/2)	100% (2/2)	0% (0/2)	0%	-
Echeverría-García et al. [34]	Etanercept	5	100% (5/5)	0% (0/5)	20% (1/5)	100% (4/4)	25% (1/4) Psoriasis worsening 25% (1/4) Erythema nodosum and vaginal infection 25% (1/4) Amniotic sac detachment
	Adalimumab	3	100% (3/3)	0% (0/3)	0% (0/3)	100% (3/3)	33.3% (1/3) High blood pressure 33.3% (1/3) respiratory distress in the newborn
	Ustekinumab	1	100% (1/1)	0% (1/1)	0% (0/1)	100% (1/1)	-
Galluzzo et al. [37]	Ustekinumab	4	100% (4/4)	-	-	100% (4/4)	25% (1/4) psoriasis worsening
Odorici et al. [36]	Adalimumab	1	100% (1/1)	0% (0/1)	0% (0/1)	100% (1/1)	100% (1/1) psoriasis worsening
	Secukinumab	2	100% (2/2)	0% (0/2)	0% (0/2)	100% (2/2)	100% (2/2) psoriasis worsening
	Etanercept	1	100% (1/1)	0% (0/1)	0% (0/1)	100% (1/1)	-
	Ustekinumab	6	66.7% (4/6)	16.7% (1/6)	16.7% (1/6)	75% (3/4) ^f^	75% (3/4) psoriasis worsening
	Infliximab	4	75% (3/4) ^g^	0% (0/4)	25% (1/4)	100% (3/3)	100% (3/3) psoriasis worsening
Kawai et al. [42]	Adalimumab	5	60% (3/5)	20% (1/5)	-	80% (4/5)	20% (1/5) Gestational hypertension and herpes simplex
Watson et al. [44]	Ustekinumab	10	100% (10/10)	20% (2/10)	0% (0/10)	100% (8/8)	12.5% (1/8) gestational diabetes
Lund & Thomsen [45]	Infliximab	6	50% (3/6)	0% (0/6)	0% (0/6)	100% (6/6)	16.7% (1/6) gestational diabetes
	Ustekinumab	4 ^f^	100% (4/4)	0% (0/4)	0% (0/4)	75% (3/4)	-
	Adalimumab	2	0% (0/2)	0% (0/2)	0% (0/2)	100% (2/2)	-
Haycraft et al. [38]	Tildrakizumab	14	71.4% (10/14) ^e^	14.3% (2/14)	28.6% (4/14)	87.5% (7/8)	12.5% (1/8) preterm labor
M. E. B. Clowse et al. [39]	Tofacitinib	16 ^d^	100% (16/16)	6.3% (1/16)	25% (4/16)	100% (9/9)	
Babuna Kobaner and Polat Ekinci [40]	Adalimumab	2	50% (1/2)		50% (1/2)	100% (1/1)	
	Etanecept	1	100% (1/1)			100% (1/1)	
	Infliximab	3	66.67% (2/3)			100% (3/3)	
	Secukinumab	1	0% (0/1)			100% (1/1)	
	Ustekinumab	2	100% (2/2)			50% (1/2)	50% (1/2) ectopic pregnancy
Russo et al. [41]	Adalimumab	2	100% (2/2)			100% (2/2)	
	Ustekinumab	1	100% (1/1)			100% (1/1)	
	Guselkumab	1	100% (1/1)			100% (1/1)	
Carman et al. [43]	Etanercept	81		16% (13/81)	8.6% (7/81)	100% (61/61)	

^a^—Women who discontinued treatment during the first or second trimester of pregnancy were considered to have dropped out of treatment. Those who were exposed to the drug until the third trimester of pregnancy did not drop out. ^b^—Spontaneous and elective abortions were eliminated. ^c^—Predominantly adalimumab and etanercept, but also other biologic drugs. ^d^—Data for 2 patients on pregnancy outcomes were lost. ^e^—Drug exposure during pregnancy was unknown for 4 patients. ^f^—One of the pregnancies was ongoing at the time of the study. ^g^—One of the women who discontinued treatment was switched to certolizumab pegol.

## Data Availability

Data are contained within the article.

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
