# Peer review of "Impact of Psoriasis and Hidradenitis Suppurativa in Pregnancy, a Systematic Review"

_jcm, 2021, doi:10.3390/jcm10245894_

Round 1

Reviewer 1 Report

In the article entitled ‘Impact of Psoriasis and Hidradenitis Suppurativa in Pregnancy, a Systematic Review authors provided a comprehensive review of pregnancy desire, relation of disease with pregnancy outcomes in 2 dermatological conditions. Although the review provides a valuable summary of the current data between 2015-2021, it does not have any synthesized data from the current manuscripts that we expect to see in a systematic review.  I have some comments.

  1. The paper lacks multiple critical elements of a systematic review. In the methods section please provide more information on the search process, how was it done? (Specify the methods used to decide whether a study met the inclusion criteria of the review, including how many reviewers screened each record and each report retrieved, whether they worked independently, and if applicable, details of automation tools used in the process)

  1. There is no information about the risk of bias of the studies and there is no statistical analysis. Present assessments of risk of bias for each included study.

  1. I am wondering if authors could provide any synthesized data from this review. If not please provide the limitations of this study also pointing out this.

  1. What is ‘Poblational cohort study’ which is extensively used in table 1? Population cohort study? ‘Nos’ in flow chart should be ‘not’ There are multiple spelling and phrasing errors.

  1. I did not see the PRISMA checklist.

Author Response

Dear Reviewer, 

Thank you for your comments, as they allow us to improve the scientific quality of our work. All the requested changes have been implemented. Moreover, below you will find a point-by-point response.

1) The search process has been explained in detail, moreover, the PRISMA checklist will be sent if the page allow us to do so. 

2) A supplementary table has been added with a detailed description of the assessment of the risk of bias for each study included in the review. It can be seen at the end of the manuscript.

3) Synthesized data have been highlighted as a final figure.

4) Spelling mistakes and expressions have been improved.

Reviewer 2 Report

This well-written article reports issues about pregnancy in 2 different dermatology diseases namely psoriasis and hidradenitis.

Only small details should be corrected:

- Page 7, lines 171-172 : acytritene : do the authors mean isotretinoin or acitretine?

- P 7: unclear about prescription of contraceptive pill after pregnancy and when there is a desire of pregnancy??

- about desire of pregnancy, it should probably be emphasized that HS patients have a poor self-image and self confidence and therefore do not dare to start a relationship, being ashamed of their body (some references may be obtained by crossing HS and sexual health, for instance Yee et al, Int J Women Dermatology, 2020

- Page 8, line 210: post ?

Line 211: postular means pustular? 

Author Response

Dear Reviewer, 

We would like to thank you for your comments, as they allow us to improve the scientific quality of our work. All the changes have been performed. Here is a point-by-point response: 

  • Spelling mistakes have been corrected.
  • The information which was confusing has been removed, as it did not provide information of interest.
  • The fact that HS patients have a poor self-image has been added, and the reference has been made.